Full Paper – MIDL 2024

# IST-editing: Infinite spatial transcriptomic editing in a generated gigapixel mouse pup

**Jiqing Wu**[1]                                                                                  JIQING.WU@USZ.CH

[1] *Department of Pathology and Molecular Pathology, University Hospital, University of Zurich, Switzerland.*

**Ingrid Berg**[1]                                                                               INGRID.BERG@BIOL.ETHZ.CH

**Viktor H. Koelzer**[1,2]                                                            VIKTOR.KOELZER@USB.CH

[2] *Institute of Medical Genetics and Pathology, University Hospital Basel, Switzerland.*

**Editors:** Accepted for publication at MIDL 2024

## Abstract

Advanced spatial transcriptomics (ST) techniques provide comprehensive insights into complex organisms across multiple scales, while simultaneously posing challenges in biomedical image analysis. The spatial co-profiling of biological tissues by gigapixel whole slide images (WSI) and gene expression arrays motivates the development of innovative and efficient algorithmic approaches. Using Generative Adversarial Nets (GAN), we introduce **I**nfinite **S**patial **T**ranscriptomic **e**diting (IST-editing) and establish gene expression-guided editing in a generated gigapixel mouse pup. Trained with patch-wise high-plex gene expression (input) and matched image data (output), IST-editing enables the seamless synthesis of arbitrarily large bioimages at inference, *e.g.*, with a $106496 \times 53248$ resolution. After feeding edited gene expression values to the trained model, we simulate cell-, tissue- and animal-level morphological transitions in the generated mouse pup. Lastly, we discuss and evaluate editing effects on interpretable morphological features. The code and generated WSIs are publicly accessible via https://github.com/CTPLab/IST-editing.

**Keywords:** Gene expression editing, spatial transcriptomics, GAN, WSI, mouse pup

## 1. Introduction

Recent advances in multi-omics technologies (*e.g.*, spatial transcriptomics (ST) (Moses and Pachter, 2022)) and generative artificial intelligence (AI) have the potential to revolutionize biomedical image analysis (Royer, 2023). Leveraging spatial co-profiling of high-plex mRNA transcripts (acting as proxies for gene expression) and high-resolution biomedical images, researchers possess unprecedented opportunities to model the complex spatial organization of an entire organism.

Concurrently, generative AI (Bermano et al., 2022; Croitoru et al., 2023) has showcased remarkable progress in creating high-quality visual content, paving the way towards novel applications in the biomedical domain. Trained with Hematoxylin and Eosin (H&E)-stained or (immuno)fluorescence images, prior studies (Carrillo-Perez et al., 2023; Lamiable et al., 2023; Wu and Koelzer, 2023) have achieved impressive results of bioimage generation and manipulation using GAN approaches. Recently, researchers (Wu and Koelzer, 2024) further demonstrated the algorithmic editability on ST data and simulated cellular morphological transitions by shifting gene expression distributions. Notably, these studies were carried out at cell- or tissue-level and the generated bioimage resolution is usually smaller than $256 \times 256$. Due to scalability limitations and violation of the translation equivariance property, the generative competence of such algorithmic methods cannot be extended to the entirety of a WSI without inducing visible stitching artifacts, which can be partially mitigated by employing more hardware resources. A series of StyleGAN studies (Karras et al., 2020, 2021) first

showed the feasibility of training $1024 \times 1024$ images on 8 V100 GPUs. In a recent paper, a GAN-based approach (Kang et al., 2023) has accomplished $4096 \times 4096$ image generation with remarkably fine details. Critically, this achievement was made possible by training a scaled GAN model on 96-128 A100 GPUs. Despite impressive breakthroughs in generating megapixel-resolution images, the hardware requirements for synthesizing WSIs at the gigapixel scale can be computationally intractable, making the model application prohibitively expensive in biomedical research.

To extend the model applicability to arbitrarily large images, Single GAN (SinGAN) (Shaham et al., 2019) and Single Denoising Diffusion Model (SinDDM) (Kulikov et al., 2023) were proposed to learn the internal statistics of a given training image. Their shared coarse-to-fine architectural design enables the generation of image samples of any desired dimensions. Differing from single-image training, InfinityGAN (Lin et al., 2022) re-introduced large-scale training on patch-wise image data using low computational resources. Tailored for high-resolution natural scene creation, strong coordinate priors, such as vertical rapid saturation and horizontal repetitive patterns of sky, land, or ocean, were imposed within the structure and texture synthesizer of InfinityGAN.

In bioimage generation, the utility of coordinate priors is nonetheless undesirable. This is because the arrangement of biological structures is not dictated by a rigid coordinate system, but rather by the intricate interplay between genetic, epigenetic, and gene expression variability that leads to the phenotype of a living system (Haniffa et al., 2021). Here, we propose **I**nfinite **S**patial **T**ranscriptomic **e**diting (IST-editing) in a generated gigapixel mouse pup. To the best of our knowledge, we are the first to introduce algorithmic gene expression editing at the scale of an entire organism:

- Taken gene expression data as the input, we achieve the seamless generation of $106496 \times 53248$ WSIs of a whole mouse pup.

- By gene expression-guided editing, we simulate cell-, tissue- and animal-level morphological transitions, measured with interpretable morphological features.

- Importantly, the model training and inference can be efficiently executed on a single consumer-grade GPU, *e.g.*, GeForce RTX 3090 Ti.

## 2. The proposed IST-editing

To efficiently process the paired transcript count array and biomedical image with matched gigapixel resolutions, we develop IST-editing upon the StyleGAN (Karras et al., 2020, 2021) framework. This is motivated by recent GAN studies (Sauer et al., 2022, 2023; Kang et al., 2023) in response to the remarkable advances made by diffusion models. While being orders of magnitude faster at inference time, these methods, built upon advanced GAN architectures such as StyleGAN, exhibit superior generation and editing performance that remain competitive with their diffusion counterparts.

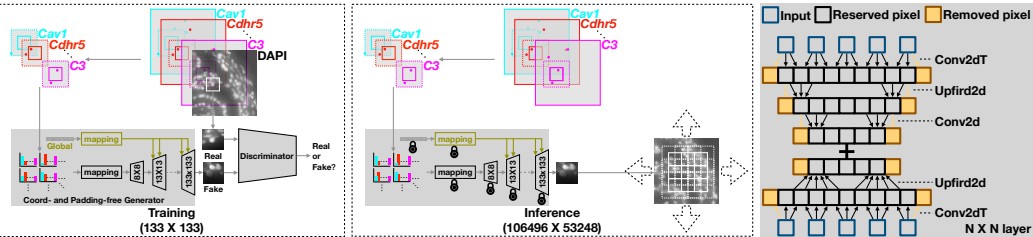

Figure 1: **Conceptual illustrations of the proposed model**.

## 2.1. ST data

**Spatial gene expression as the input and representation**: In the natural image domain, previous generative models (Shaham et al., 2019; Kulikov et al., 2023) typically utilize (spatial) noise input for unconditional image generation. In addition, learned textural representations (Radford et al., 2021) can be incorporated into the model to guide the image alterations (Bermano et al., 2022). However, semantic ambiguity often occurs in interpreting a single latent code and qualitative analysis is mostly made possible for a subset of representations (Härkönen et al., 2020). Given the well-established biological understanding of many individual genes, we utilize gene expression as both the **input data** and **interpretable representation** for bioimage generation and editing.

**Training data pair**: As shown in Fig. 1 (left), we take the patch-wise spatial gene expression (input) and biomedical image (output) as the training data pair. During the training, we randomly and densely crop $2n \times 2n$ gene expression arrays that are center-aligned on the paired $n \times n$ image. With a $2\times$ higher resolution than the associated image, these gene arrays will eventually allow the construction of a spatial grid that imposes seamless WSI generation at inference. To strike a balance between the generation quality and training efficiency, we employ the paired $256 \times 256$ gene array and $128 \times 128$ image in the experiments. To ensure boundary consistency between neighboring generated images, every boundary pixel at $(x, y)$ of the image tile is obtained using gene expression values located at $(x', y')$, where $|x' - x| \leq 64$ and $|y' - y| \leq 64$. A larger $256 \times 256$ gene array, containing all these values, is thus necessary to generate a $128 \times 128$ image tile. Due to the sparse spatial presence of gene expression, we down-scale the sampled gene array to $8 \times 8$ by sum reduction, such that more densely distributed gene expression values are aggregated in the format of a smaller 3D array.

## 2.2. Training

**Coordinate- and padding-free generator** $G$: Instead of relying on strong coordinate-based priors including the vertical saturation and horizontal repetition of natural scenes, the design of our generator is driven by the intricate interaction between genes (causative factors) and phenotypes (observable characteristics). To model the directed linkage from gene expression to the biomedical image, we propose a straightforward coordinate-free generator, which is constructed using a series of padding-free and translation equivariant StyledConv layers (Fig. 1). No external prior knowledge, aside from gene expression data, is incorporated into the output images. In all padding-free layers, we discard pixel values that are padded at both spatial ends of the output. Consider $i = 1, 2, \ldots, l$, we then have the intermediate output with $(2^{i+2} + 5) \times (2^{i+2} + 5)$ spatial resolutions for the $i$-th layer. After discarding 5 boundary pixels of the last layer output, we obtain the generated patch-wise image. Leveraging the consistent $2\times$ increase of image resolution, our model can be easily adapted to output patches with $256 \times 256$ or $512 \times 512$ resolutions.

**Cell-subtype conditioned discriminator** $D$: Inspired by conditional generations of well-characterized normal and cancer cellular images (Wu and Koelzer, 2024), we integrate cell subtype information into the discriminator to adversarially and conditionally train the generator. Concretely, we project cell label embeddings into $D$ and train both models with the conditional adversarial loss $\mathcal{L}_{\mathsf{adc}}$. Along with the $R_1$ regulation $\mathcal{L}_{R_1}$ and path length regulation loss $\mathcal{L}_{\mathsf{path}}$ (Karras et al., 2020), we have the loss function $\min_G((\max_D \mathcal{L}_{\mathsf{adc}}) + \alpha_{R_1} \mathcal{L}_{R_1} + \alpha_{\mathsf{path}} \mathcal{L}_{\mathsf{path}})$, where $\alpha_{R_1}$ and $\alpha_{\mathsf{path}}$ are hyperparameters and are determined to be 10 and 2 based on the prior study (Wu and Koelzer, 2024). Then, we train the GAN model for 800k iterations with a batch size of 16. Eventually, the optimal model performance is determined using Fréchet Inception Distance ($d_{\mathsf{FID}}$) (Heusel et al., 2017) and high

Peak Signal-to-Noise Ratio (PSNR). For the former, we use a more efficient implementation (Wu and Koelzer, 2022) and robust CLIP features (Radford et al., 2021; Kynkäänniemi et al., 2022) to carry out the computation.

### 2.3. Inference

**Spatial gene expression grid**: We employ a divide-and-conquer strategy at inference, breaking down the WSI generation into parallelizable subtasks of patch-wise image generation. To guarantee the boundary consistency of neighboring patches and as shown in Fig. 1 (middle), we use the spatial gene expression grid (dotted lines) that is overlaid on the image grid (solid lines). This grid is formed and merged with $2n \times 2n$ gene expression arrays center-aligned on $n \times n$ images, in which the stride size of array shift is $n \times n$. Together with the padding-free layer design (Fig. 1 (right)), we generate arbitrarily large WSIs given gene expression data as the input. Using a single GeForce RTX 3090 Ti, it takes $\sim$ 30 mins to synthesize $106496 \times 53248$ WSIs, which are accessible via our GitHub repo and can be thoroughly examined by open-source software such as QuPath (Bankhead et al., 2017).

## 3. Experiments

We test IST-editing on the public Xenium (Janesick et al., 2022) ST dataset of a one-day mouse pup. This gigapixel-resolution dataset[1] provides a well-curated sparse 3D array of 379-plex gene transcript counts (App. Fig. 4) and the spatially matched DAPI-stained WSI at the identical resolution, offering a comprehensive morpho-molecular landscape of the whole organism. In the absence of clear cell-level annotations in the Xenium dataset, we conducted a careful evaluation of the WSI and cell-level clustering[2] within the context of tissue organizations. Instructed by domain biomedical experts, we confirmed the accuracy of subtype assignments derived from the 'kmeans_10_clusters' results in the raw data. Depending on the majority vote of cell subtypes presented in the sampled training data, we assign the label of the predominant subtype to each image tile.

### 3.1. Evaluation of generation results

We benchmark IST-editing against state-of-the-art diffusion- and GAN-based models such as Infinity-GAN. Consistent with the IST-editing approach, we feed all the models with patch-wise spatial gene expression data (input) and DAPI images (output) for systematic and fair comparisons. Following the single-image training paradigm, we train SinGAN (Shaham et al., 2019) and SinDDM (Kulikov et al., 2023) on individual tissue-level images (*e.g.*, $4096 \times 4096$) and generate high-resolution images for direct comparison with the IST-editing results. In contrast, StyleGAN2, InfinityGAN, and IST-editing are trained on patch-wise data pairs extracted from the entire WSI. As shown in Fig. 2 (a), SinGAN and SinDDM can recreate low-resolution images (small inset, left) including texture similarities to the original tissue such as the alveolar pattern observed in samples from the lung region. However, the image generation cannot be consistently scaled to a higher resolution: Only basic and biologically meaningless tissue textures remain. StyleGAN2 preserved a pattern resembling cell nuclei in generated high-resolution images. Nevertheless, the tissue structure corresponding to

---

1. The download link is https://s3-us-west-2.amazonaws.com/10x.files/samples/xenium/1.6.0/Xenium_V1_mouse_pup/Xenium_V1_mouse_pup_outs.zip

2. Please see also the 10x Genomics data summary provided at https://cf.10xgenomics.com/samples/xenium/1.6.0/Xenium_V1_mouse_pup/Xenium_V1_mouse_pup_analysis_summary.html.

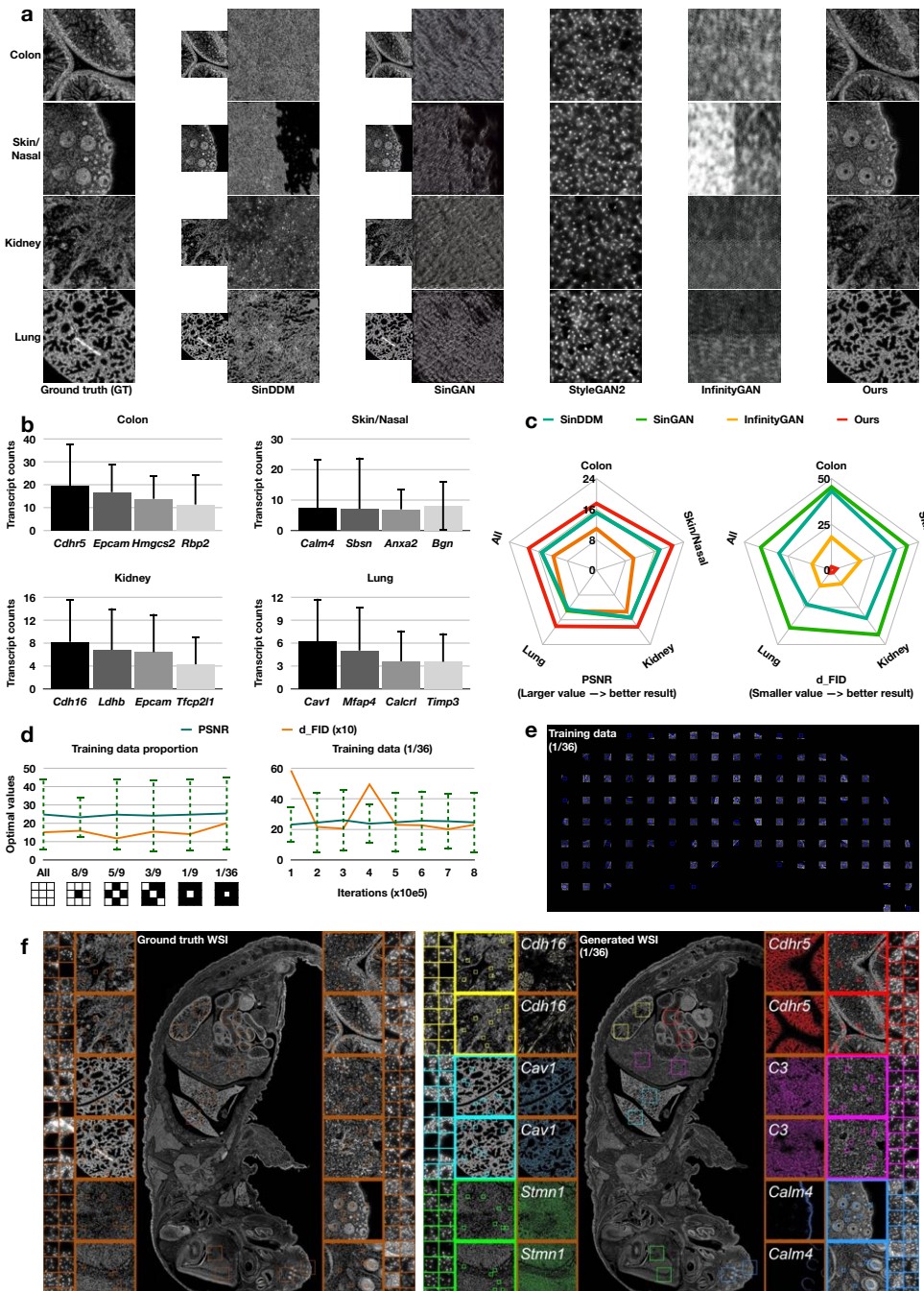

Figure 2: **Experimental results of the WSI generation**. **a**. The visual comparison of tissue-level (4096×4096) synthesized images obtained by training with 100% of the available data using SinGAN (Shaham et al., 2019), SinDDM (Kulikov et al., 2023), StyleGAN2 (Karras et al., 2020) and InfinityGAN (Lin et al., 2022) as compared to IST-editing. Using the coarse-to-fine upscaling technique introduced by SinGAN and SinDDM, we present unsatisfactory upscaling results (4096 × 4096, right plots) next to the faithful generation of low-resolution input images (left subplots) for a clear side-by-side comparison. **b**. The mean and standard deviation of transcript counts of the highly expressed genes (per cell) *w.r.t.* individual tissue regions. **c**. The comparison of tissue-level generation results between the compared methods by PSNR (left) and $d_{FID}$ (right). **d**. The comparison of PSNR and $d_{FID}$ scores obtained by training IST-editing on progressively smaller subsets of available data (left) and at different numbers of iterations for training with 3% of the available data (right). For these experiments, subsets of the available data are sampled following the 'checkerboard' patterns, as illustrated underneath the 'Training data proportion' plot. **e**. The visual illustration of 3% of available training data. **f**. The cell-, tissue- and animal-level visualization of ground-truth (left) and generated (right) mouse pup WSI. To visualize the spatial pattern of leading gene expression in the right plot, we first downscale the resolution of gene expression array using sum reduction and then shift the gene expression level to [0, 255].

the individual organ regions is lost, as is evident from the 'StyleGAN2' column of Fig. 2 (a). Owing to undesired coordinate priors for bioimage generation, we observed horizontal lines and repetitive patterns in images generated by InfinityGAN and clearly identifiable tissue structures are not present in these image examples. After inputting the 379-plex gene expression data (*e.g.*, see Fig. 2 (b) and App. Fig. 4), our approach successfully generates tissue-level images at the scale of $4096 \times 4096$ resolution, with biologically meaningful details (Fig. 2 (a), right). The generated images show a high level of similarity both in tissue organization, texture and cell-level detail to the biological prior, as supported by expert pathologist interpretation. Quantitatively illustrated in Fig. 2 (c), IST-editing outperforms compared methods in terms of low $d_{\mathsf{FID}}$ and high PSNR score. Using the padding-free StyledConv operations and spatial gene expression grid, IST-editing achieved the WSI generation with a $106496 \times 53248$ pixel resolution. Please see also App. Fig. 5 for more elaborated visualization.

**Training data utility (100% - 3%)**: Next, we evaluate the generation robustness of the proposed approach under conditions of increasing data scarcity. For this purpose, we utilize progressively smaller subsets of the available data for training. As depicted in Fig. 2 (d, left), the optimal $d_{\mathsf{FID}}$ and PSNR scores remain consistent as the amount of available data decreases. Only when reducing the training data to 1/36 of the original size (Fig. 2 (e)) do we start to observe a mild degradation in quantitative performance by $d_{\mathsf{FID}}$. Upon comparing the cell-, tissue- and animal-level generation quality achieved by training on the entire dataset (App. Fig. 5 (a, b)) and 3% (Fig. 2 (f, right)) of the available data, the visual discrepancy between the two gigapixel-resolution WSIs appears marginal, substantiating the adaptability of IST-editing to limited data scenarios, requiring the seamless synthesis of more than 97% of the unseen data.

## 3.2. Evaluation of editing effects

We investigate gene expression-guided editing of WSI data by three distinct strategies. Experiments are performed on the generated 'in-silico mouse pup' which contains co-profiled ST and WSI data of all major mammalian organ systems.

**(1) Direct scaling of gene expression**: Organized structures of diverse tissue regions emerge when progressively scaling the expression levels of the top four genes by a factor of 0.5, 1 (baseline), and 2 (Fig. 3 (b, left and middle)), while remaining gene expression values are zeroed out. Such targeted editing is driven by the observable dominant impact on the morphological generation of the top four leading expressed genes (Fig. 2 (b)). Interestingly, the editing effects exhibited biologically explainable heterogeneity across the different regions. In the colon section, we observe the emergence of crypt epithelial structures orchestrated by the upscaling of leading genes including Epithelial Cell Adhesion Molecule, *EPCAM*. As muscle-specific genes are not represented in the top colon gene sets, the outer muscle layer remains absent in the reconstruction. In other examples, the clear structures and organizations of the lung region have been recovered by our approach, closely resembling the GT lung image, and image artifacts (*e.g.*, white fluff on GT WSI scan) are effectively eliminated in the reconstruction. Calculated on the proportional ratio between edited and GT tissue regions highlighted in the bounding boxes (Fig. 3 (d)), the radar charts in Fig. 3 (b, right) demonstrate a consistent increase in cell-level metrics approaching the GT with the up-scaling coefficients.

**(2) Indirect scaling of gene expression**: Similar to the cell-level manipulation study (Wu and Koelzer, 2024), we perform algorithmic editing on the sample covariance matrix (SCM) and scale the leading eigenvalues by 0.1, 0.5, and 1 (baseline). Consider the SCM $\frac{1}{n}G^{\mathsf{T}}G = O\lambda O^{\mathsf{T}}$, where $G$ is the collection of $n$ 379-plex gene expression data from a given tissue region, $O_i$ is the $379 \times 379$

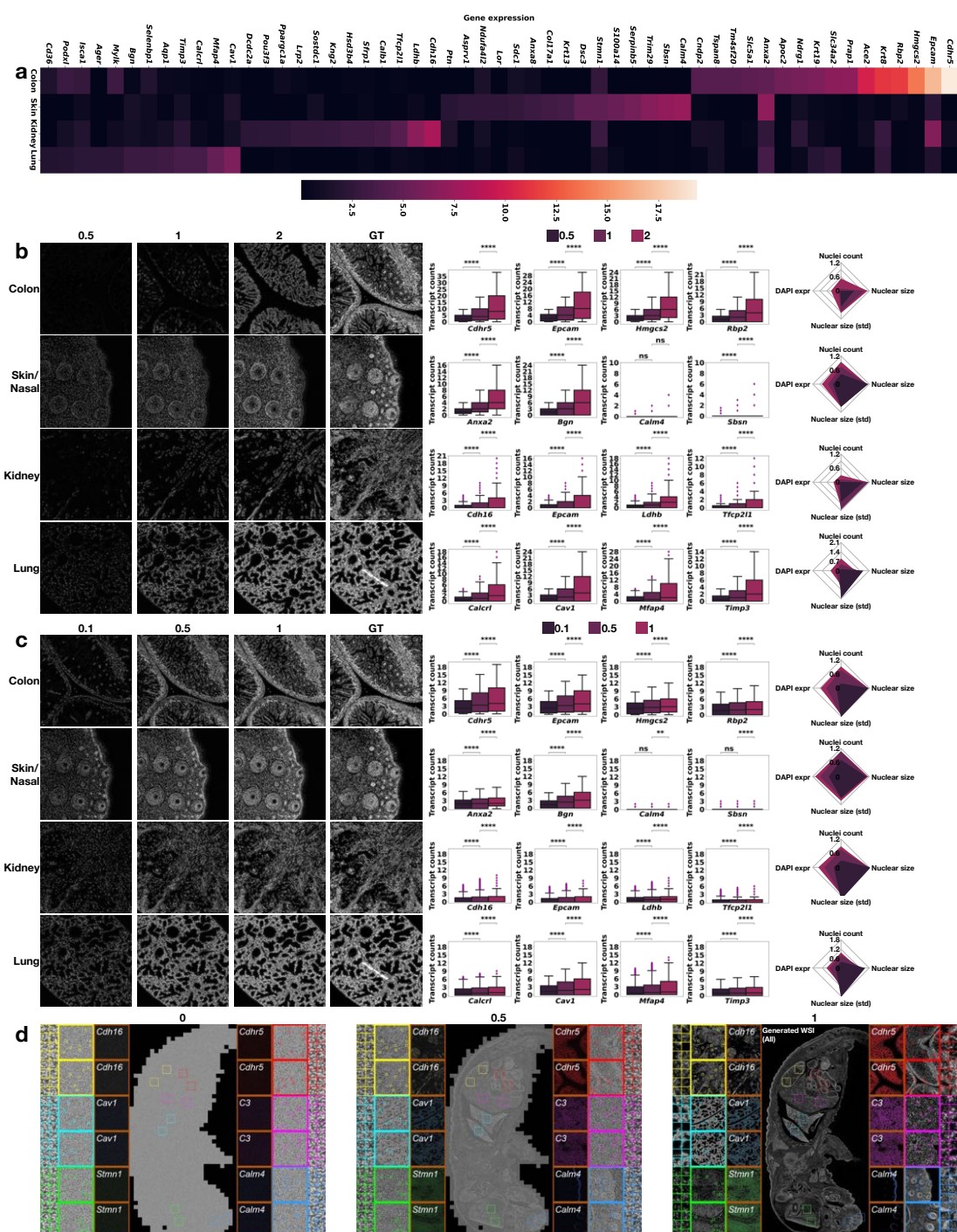

Figure 3: **Gene expression profiles and experimental results of IST-editing**. **a**. The heatmap of highly expressed genes (average per cell) *w.r.t.* different tissue regions of the whole mouse pup and selected organ systems of interest. **b**. The visual (left) and quantitative (right) IST-editing effects on individual tissue regions obtained by scaling the leading gene expression group (middle) while zeroing out the rest of gene expression values. **c**. The visual (left) and quantitative (right) IST-editing effects on individual tissue regions obtained by scaling the leading eigenvalues of the sample covariance matrix (SCM) ([Wu and Koelzer](), [2022](), [2024]()). For both (b) and (c), **** means $p \leq 0.0001$ and 'ns' stands for not statistically significant. The error bar of the box plot represents the 5%–95% quantile. The radar plots report the proportional ratio of morphological features between edited (numerator) and GT cells (denominator). **d**. The overall editing effects on the whole mouse pup achieved by the interpolation between random noise and ground-truth gene expression values. For plots (b)-(d), all the editing experiments are conducted using the model trained with 100% of the available data.

eigenbasis and $\lambda$ is the (sorted) diagonal eigenvalues derived from eigenvalue decomposition. Then, we control $\lambda$ for indirectly conducting gene expression-guided editing. As illustrated in Fig. 3 (c), there exists a rather homogeneous transition of tissue structures across the various regions of interest. On the contrary to the results described above using the leading genes, the muscle layer of the colon tissue as well as global architectural features of lung and skin are already observed at the scale of 0.1 when using all genes as an input. After examining the editing effects with up-scaling of the eigenvalues, we witness a further increase in DAPI pixel intensity and increased sharpening of architectural details closely resembling the GT image. This is reflected by the quantitative analysis of the interpretable morphological features Fig. 3 (c, right), where we observe an expected increase in the cellular region and DAPI signals.

**(3) Interpolation between unorganized and well-organized gene expression**: To simulate morphological transitions at the scale of a whole 'in-silico mouse pup', we conduct linear interpolation between randomly sampled and ground truth spatial gene expression, generating WSI results at coefficients of 0 (noise), 0.5, and 1 (mouse pup). The resulting WSIs exhibit a gradual progression from chaotic cellular organization - as reflected through the appearance of 'random noise' across the entire sample - to the highly organized structure of the one-day mouse pup. We thus demonstrate the versatility of IST-editing in simulating biological processes across multiple scales.

### 3.3. Evaluation of model limitations

**Training data utility (0.1%)**: Pushing the limits further, we conduct extreme stress tests on the proposed approach for reconstructing the whole mouse pup. This is carried out by training on a single 2048×2048 resolution image extracted from individual tissue regions such as kidney, lung, and brain. Though the overall outline and structure of the mouse pup are retained, IST-editing struggles to recreate the WSI with fine biological-aware details, as illustrated in App. Fig. 7. Remarkably, heterogeneous generation patterns for different organs arise when training solely on one single image. For instance, the training of the gut region image leads to the generation of blank space in the mouse brain. This can be explained by the non-overlapping highly expressed genes between the gut (*e.g.*, *Cdh16*, *Ldhb*, *Epcam*, *Tfcp2l1*) and brain (*e.g.*, *Stmn1*, *Gap43*, *Nnat*, *Tubb3*) region, as presented in Fig. 3 (a) and App. Fig. 4. When utilizing an 'almost black' image with a mere fragment of mouse skin (App. Fig. 7), the overall structure of the mouse pup remains preserved, though the cellular and tissue generation tends to exhibit a preference for mimicking skin epithelial morphology, suggesting a bias towards replicating trained cellular subtypes. To resolve the limitation of training and testing on the same WSI of the mouse pup, we reported generalization results on different brain sections from two mice (App. Fig. 9, 10) using the synergistic ST data of mouse brain atlas (Yao et al., 2023).

### 4. Discussion and Conclusion

This proof-of-concept study showcased the generative ability and editability in an in-silico mouse pup with DAPI nuclear staining and linked ST data. Notably, IST-editing can be readily extended to other broadly established staining techniques to visualize cellular detail, as exemplified by the first H&E generation results of the same mouse pup (App. Fig. 8). In-silico modeling holds great potential for the Replacement, Reduction, and Refinement of animal research and extends beyond animal modeling. In future applications, IST-editing could enable the simulated intervention on biological samples from human pathology with reduced ethical, legal, and regulatory risks and provide a novel perspective to investigate the linkage between genotype and phenotype in human diseases.

## Author contributions statement

J.W. and V.H.K. conceived the research idea. J.W. implemented the algorithm and carried out the experiments. J.W., I.B. and V.H.K. analyzed the results. J.W. and V.H.K. drafted the manuscript. I.B. critically reviewed the manuscript and supplied biological interpretations. V.H.K. supervised the project.

## Competing interests

J.W. declares no competing interests. V.H.K. declares project-based research funding from Roche and the Image Analysis Group outside to the submitted work. V.H.K. is on an advisory board of Takeda has served as an invited speaker on behalf of Indica Labs and for Sharing Progress in Cancer Care, an independent nonprofit organization, outside of the submitted work.

## Acknowledgments

This study is funded by core funding of the University of Zurich to the Computational and Translational Pathology Lab led by V.H.K. at the Department of Pathology and Molecular Pathology, University Hospital and University of Zurich.

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

## Appendix A. The overall gene expression profiles

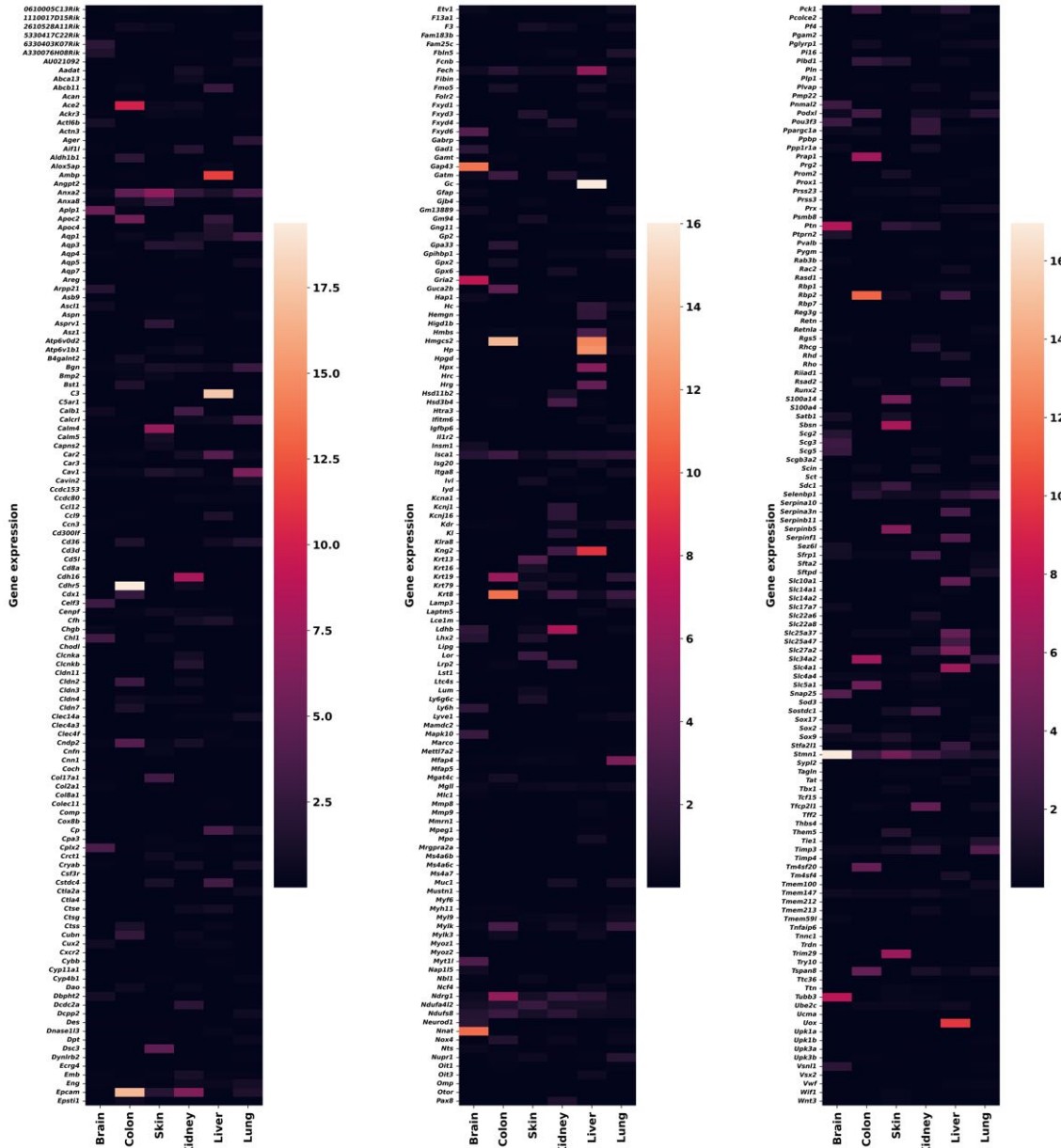

Figure 4: **The heatmap of 379-plex gene expression values (average per cell)** *w.r.t.* **different tissue regions**.

## Appendix B.  The experimental results for the DAPI-stained WSI

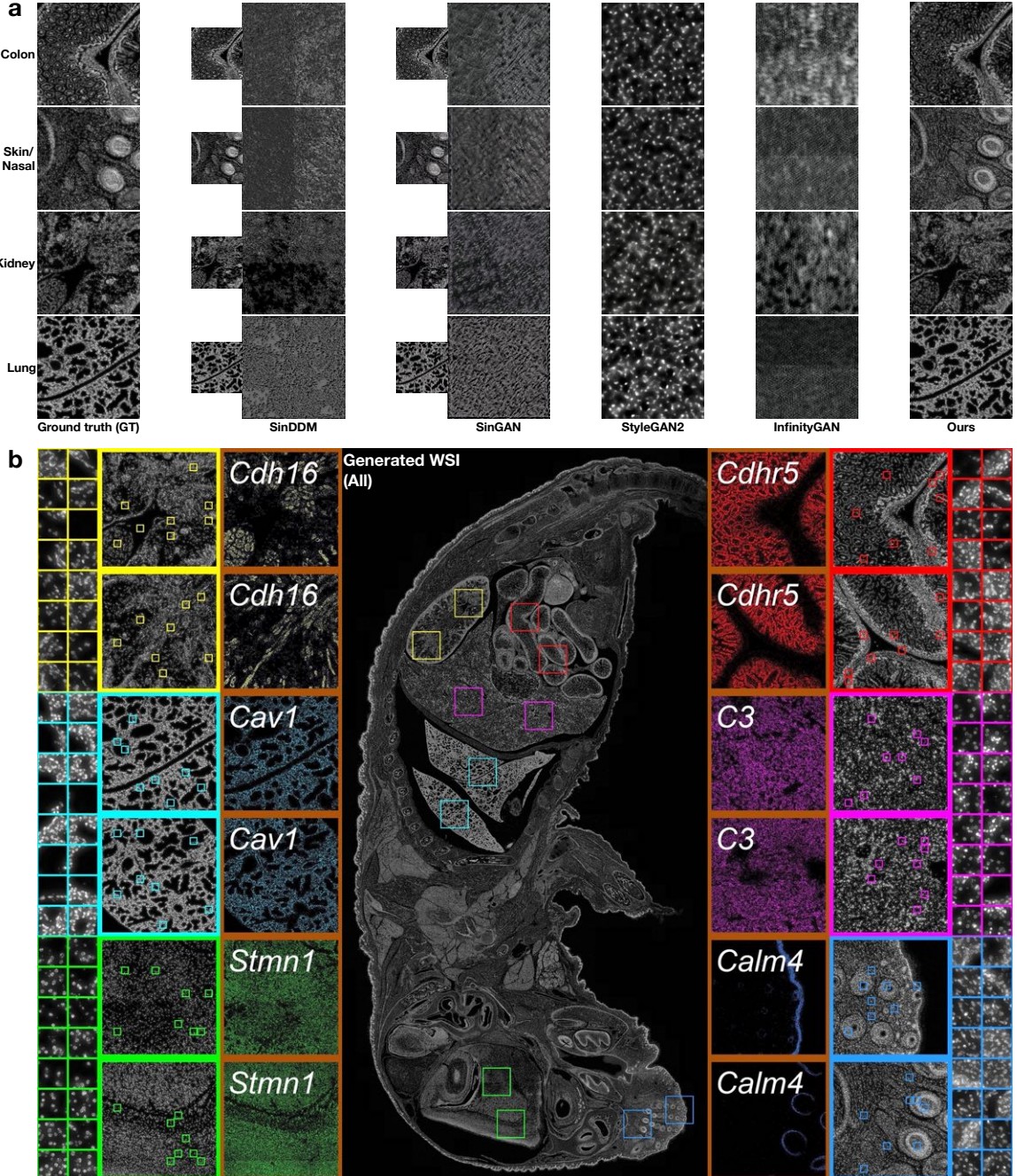

Figure 5: **The generation results of tissue region images (a) and WSI (b)**.

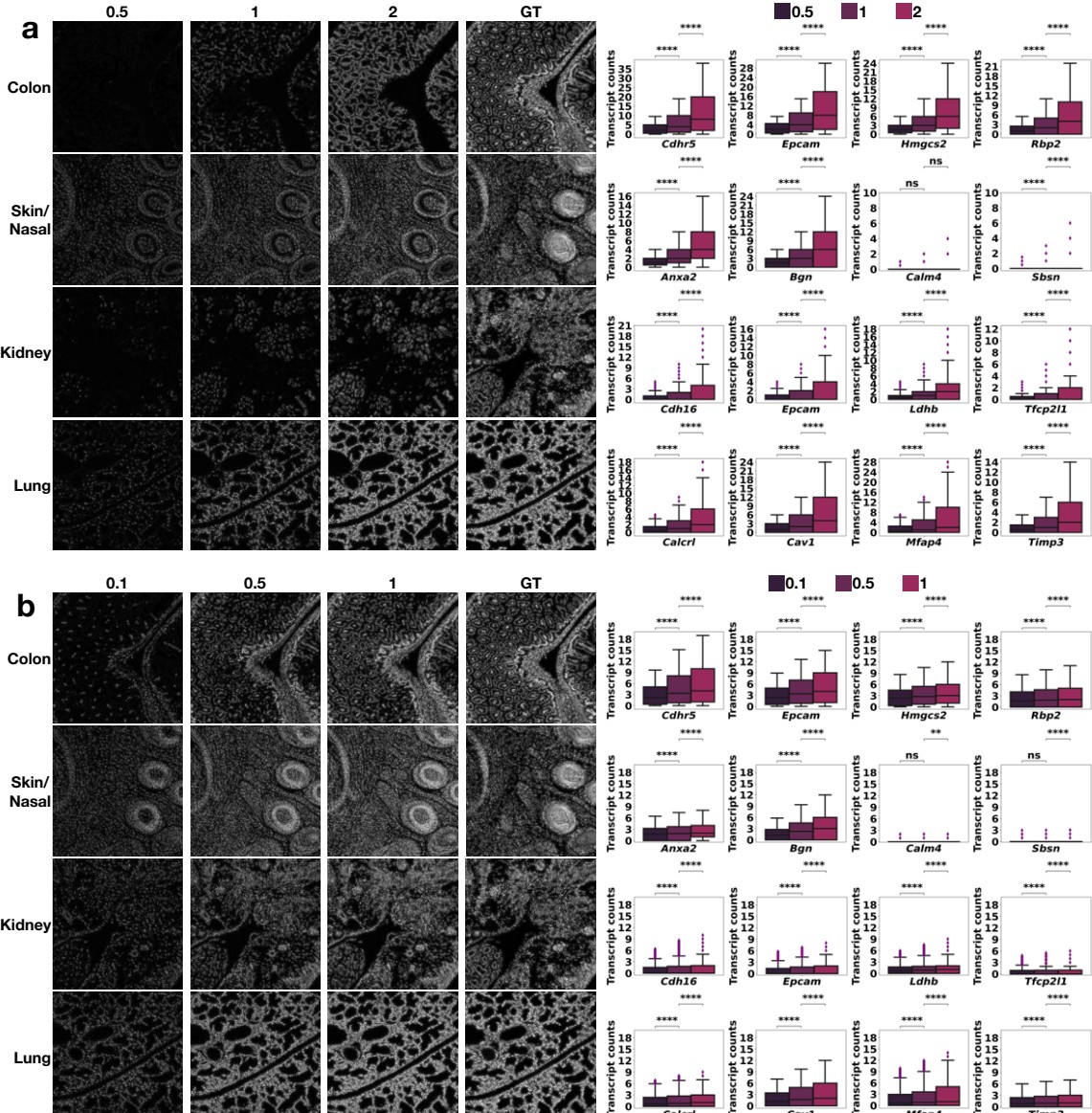

Figure 6: **The experimental results of diverse editing effects**. **a**. The visual (left) and quantitative (right) editing effects on various tissue regions by scaling the leading gene expression group (middle) while zeroing out the rest of gene expression values. **b**. The visual (left) and quantitative (right) editing effects by scaling the leading eigenvalues of the sample covariance matrix (SCM) of individual tissue regions.

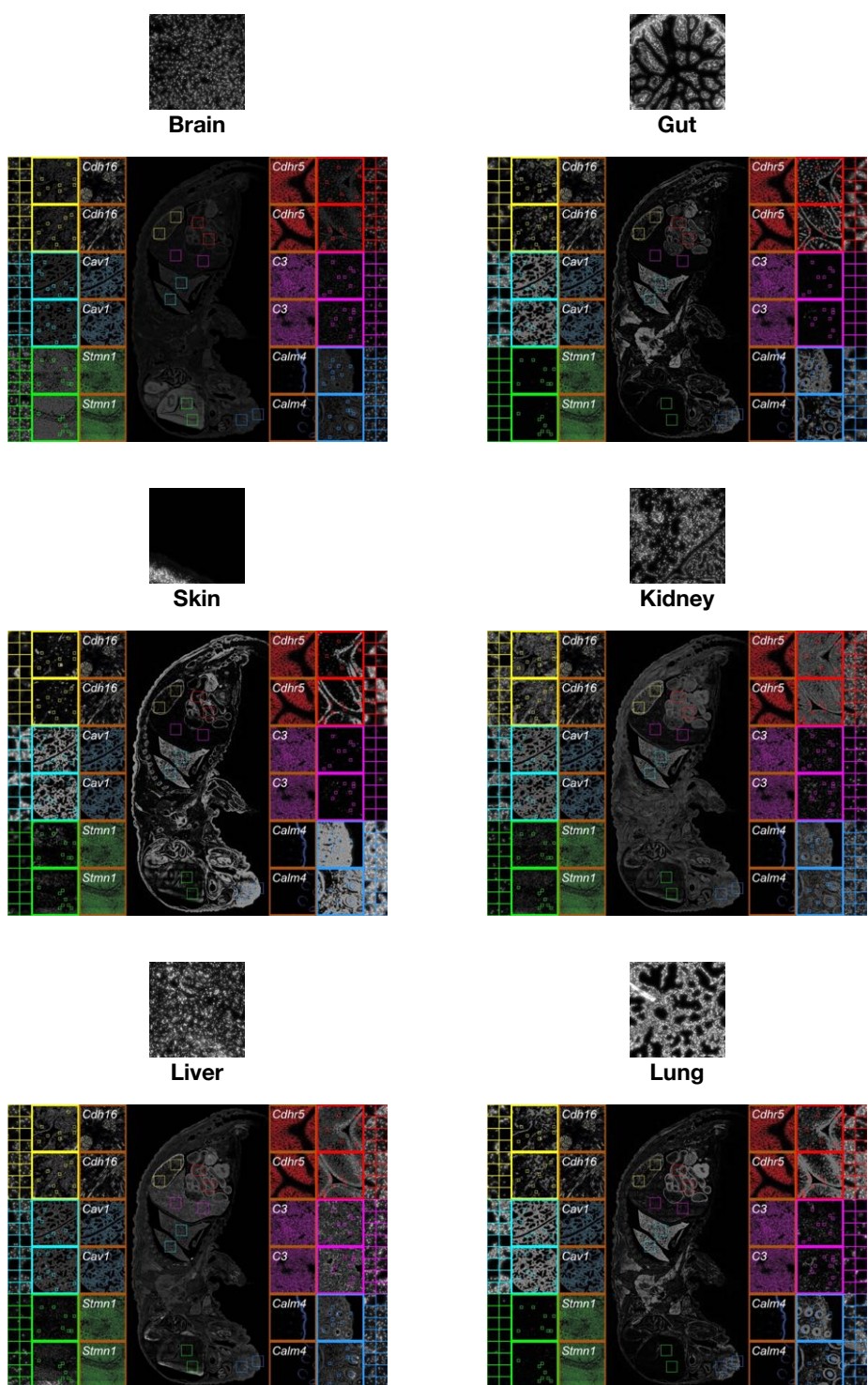

Figure 7: **The failure cases of WSI generation brought by training on a single 2048 × 2048 image extracted from individual tissue regions**.

## Appendix C. The first results for the H&E-stained WSI

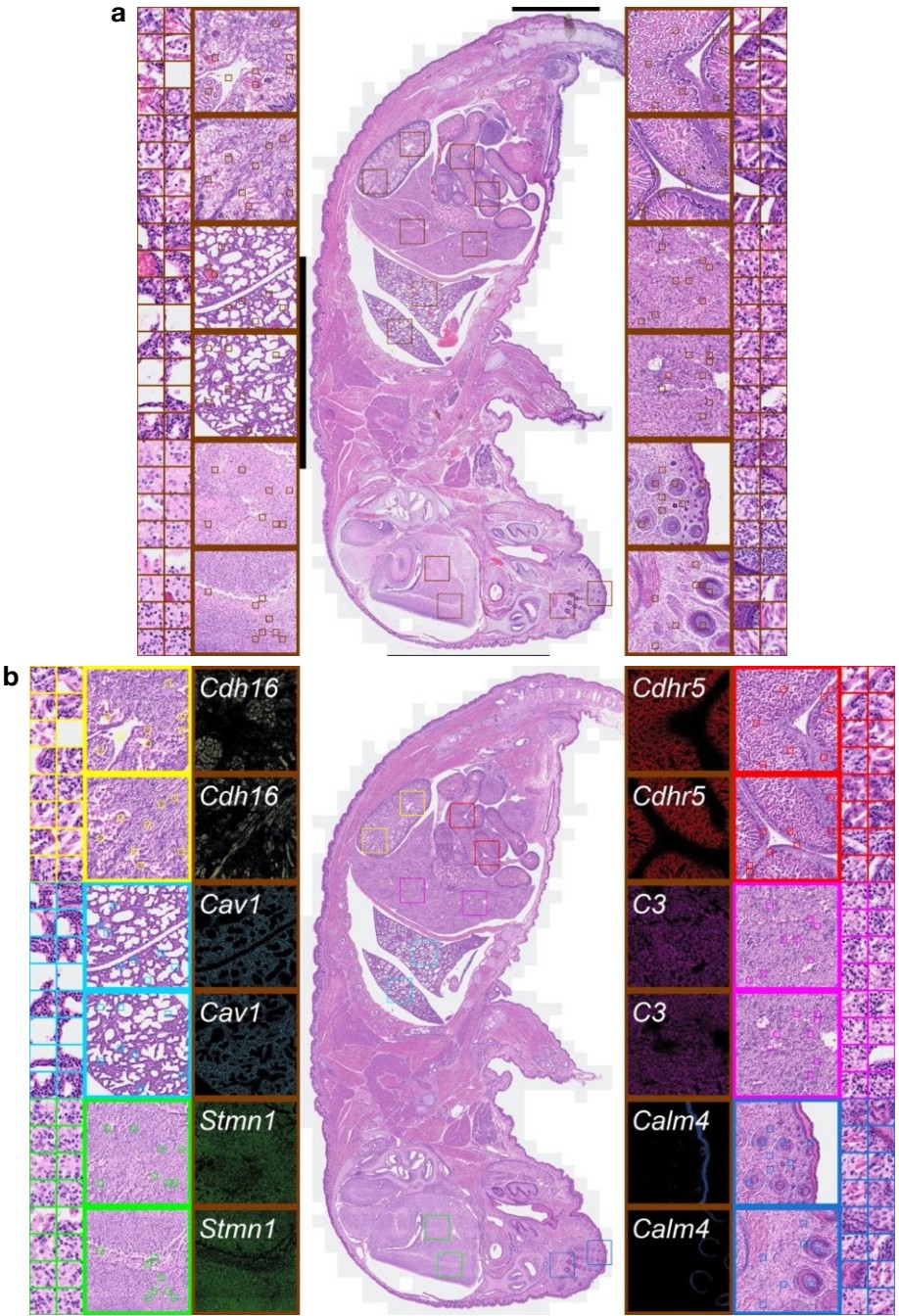

Figure 8: **The ground truth (a) and generated (b) H&E-stained WSIs**.

### Appendix D. The generalization results for coronal brain sections ([Yao et al., 2023](#))

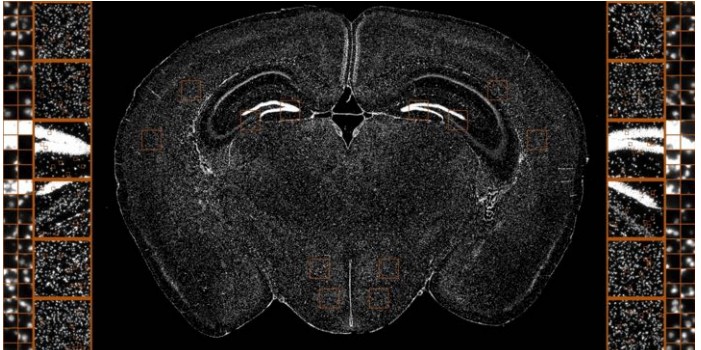

|  | Training | Test |
|---|---|---|
| PSNR (mean) | 27.64 | 27.53 |
| PSNR (std) | 9.94 | 9.64 |
| d_FID (mean) | 3.3 | 2.27 |
| d_FID (std) | 0.001 | 0.002 |

**Groundtruth for training**          **Quantitative metric**

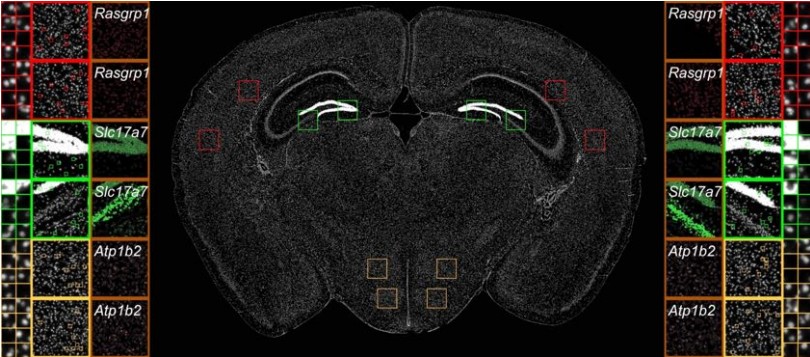

**Generation on trained data**

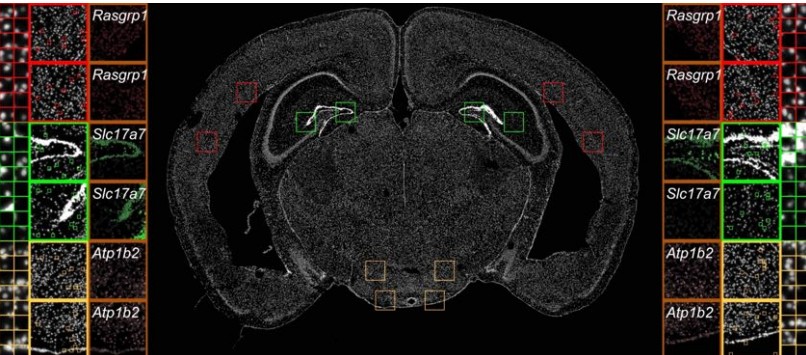

**Generation on unseen test data**

Figure 9: **The ground truth (a) and generation result of the trained (b) and test (c) WSIs**. Here, the training brain section comes from the female mouse (ID: 609882, file: 1198980117) and the test section comes from the male mouse (ID: 609889, file: 1198980478), where both ST datasets have been generated with the same gene panel. By tile-wise quantitatively comparing the generated and ground truth WSIs, we report the mean and standard deviation of PSNR and $d_{\mathsf{FID}}$ for both training and unseen test data. Same as the results reported in the main manuscript, we here use a more efficient implementation ([Wu and Koelzer, 2022](#)) and robust CLIP features ([Radford et al., 2021](#); [Kynkäänniemi et al., 2022](#)) to carry out the $d_{\mathsf{FID}}$ computation. To visualize the spatial pattern of leading gene expression values in the middle and bottom plots, we first downscale the resolution of the gene expression array using sum reduction and then shift the gene expression level to [0, 255].

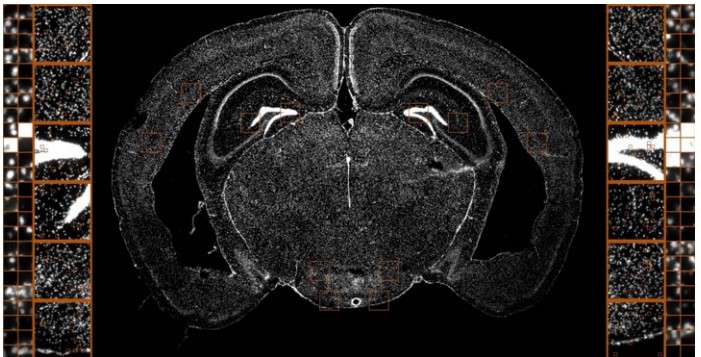

|  | Training | Test |
|---|---|---|
| PSNR (mean) | 27.37 | 21.77 |
| PSNR (std) | 9.76 | 5.63 |
| d_FID (mean) | 1.76 | 7.56 |
| d_FID (std) | 0.0002 | 0.005 |

**Groundtruth for training**          **Quantitative metric**

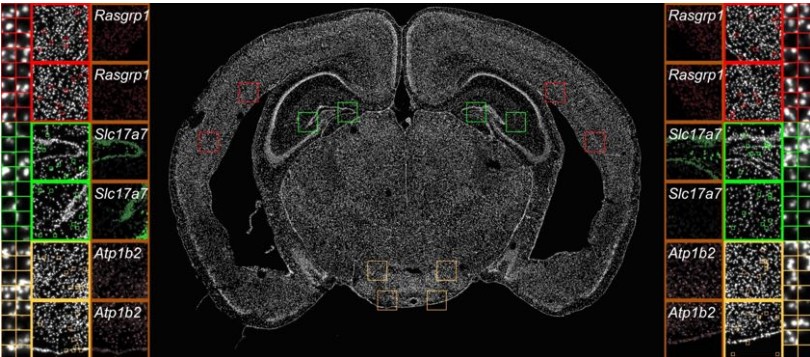

**Generation on trained data**

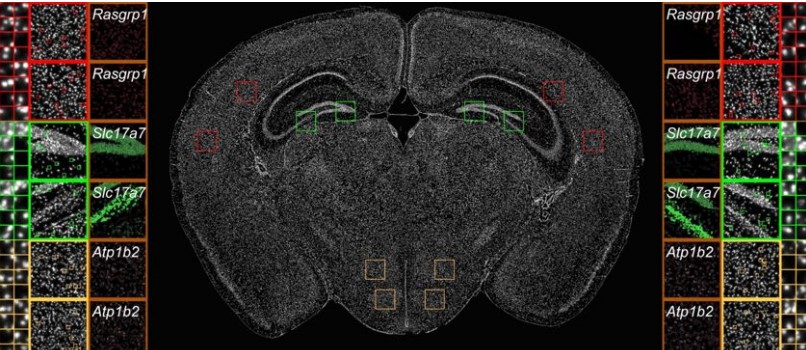

**Generation on unseen test data**

Figure 10: **The ground truth (a) and generation result of the trained (b) and test (c) WSIs**. Here, the training brain section comes from the male mouse (ID: 609889, file: 1198980478) and the test section comes from the female mouse (ID: 609882, file: 1198980117). By tile-wise quantitatively comparing the generated and ground truth WSIs, we report the mean and standard deviation of PSNR and $d_{FID}$ for both training and unseen test data. Same as the results reported in the main manuscript, we here use a more efficient implementation (Wu and Koelzer, 2022) and robust CLIP features (Radford et al., 2021; Kynkäänniemi et al., 2022) to carry out the $d_{FID}$ computation. To visualize the spatial pattern of leading gene expression values in the middle and bottom plots, we first downscale the resolution of the gene expression array using sum reduction and then shift the gene expression level to [0, 255].

