# OpenReview forum: "IST-editing: Infinite spatial transcriptomic editing in a generated gigapixel mouse pup"
_MIDL.io/2024/Conference — MIDL 2024 Oral_

### Official Review · Reviewer_HjWd · 2024-02-20

**Confidence:** 2
**Preliminary Rating:** 3
**Recommendation:** Poster
**Final Rating:** 4

**Summary:**

The paper is focused on gigapixel image synthesis conditioned by spatial gene expressions. Concretely, the authors focus on using GAN-based methods, which enable faster inference times than Diffusion models. Authors focus their experiments on mouse data (patch, tissue, organ, whole animal), and show the effectiveness of the GAN-based approach. To further motivate the applicability of such methodology, authors include several experiments on style editing during inference.

**Strengths:**

- The problem is well-motivated.
- The paper is well-structured and is easy to follow.
- The proposed GAN-based approach shows promising results better than diffusion-based generative models for the objective task.
- The solution is relatively more computationally efficient than diffusion-based approaches, which is positive considering the large scale of WSIs.
- The proposed editing effects properly justify the motivation of the work, as a proof-of-concept of the potential of applying generative models transcriptomic and histology data.

**Weaknesses:**

- The differences w.r.t prior relevant literature (e.g. Carrillo-Perez 2023) need to be more clearly established. Why did the authors exclude their proposed RNA-GAN as a baseline?
- As stated in the manuscript, Kang et al., 2023 have recently revised GAN architectures as an efficient alternative to diffusion model-based image synthesis. Nevertheless, they showed that the latter still produces a better-conditioned generation in most cases. Nevertheless, this does not seem to be the case in this work. For instance, DDM shows largely worse performance than the GAN-based approach. Do the authors find any explanation for this? Since SinDDM implementation details are scarcely detailed, this raises concerns about the proper adjustment of the baselines.
- As far as I understand, Xenium dataset only contains one slide of one unique study. Is this the case? Please, clarify otherwise the train/eval/test partitions used. If Xenium only consists of one unique slide, how can authors prosperity evaluate the generalization capabilities of the proposed approach?
- Some experimental details are not specified. For instance, how are alpha hyperparameters in Eq. 1 fixed? To which values?

**Detailed Comments:**

No additional comments.

**Justification Of Final Rating:**

The authors have satisfactorily addressed my concerns during the rebuttal regarding baselines, implementation details, and generalization experiments. The authors have included such clarifications in the manuscript.

**Justification Of The Preliminary Rating:**

The paper's scope and experiments are well-motivated. Nevertheless, I have some concerns regarding the experimental setting and baselines. I tend toward increasing my score if those are properly justified.

**Questions To Address In The Rebuttal:**

Please, see Weaknesses.

**Special Issue:**

No

---

> ### Author Response · Authors · 2024-03-16
> **To Reviewer HjWd**
>
> Thank you for your approval of our motivation, promising results, and computational efficiency.
>
> **Q1:** Why did the authors exclude RNA-GAN as a baseline?
>
> **A: (Translation equivariance)** Thank you for this important question. The rationale behind the exclusion is that the architectural variants of RNA-GAN violate the translation equivariant property. As a result, RNA-GAN cannot generate arbitrarily large WSIs without introducing tile stitching artifacts.
>
> We have supplied the theoretical argument in the revision.
>
> ____
>
> **Q2:** Do the authors find any explanation for the largely worse performance by SinDDM?
>
> **A: (Upscaling challenge)** Following 1) the single-image training scheme at a lower resolution (e.g., 223 X 223) and 2) the coarse-to-fine upscaling trick introduced by SinDDM, the worse performance comes from the upscaling challenge from 223 X 223 to 4096 X 4096.
>
> The original SinDDM code can be found at https://github.com/fallenshock/SinDDM. In the experiments, we follow default hyperparameters configured in the repo, and only changes in integrating gene array input have been made.
> Eventually, you can access our revised [code](https://drive.google.com/file/d/1GbjDB9lnqNhwah1VUsTDMT0pxQ852SIy/view?usp=sharing) for a side-by-side implementation comparison.
>
> ____
>
> **Q3:** If Xenium only consists of one unique slide, how can authors prosperity evaluate the generalization capabilities of the proposed approach?
>
> **A: (Newly released data on Dec. 2023)** You are right, we indeed struggled to evaluate the model generalization due to the rare availability of ST datasets at sub-cellular resolution.
>
> Thanks to newly released ST data [1] of the mouse brain, we can now train and test on different brain sections from two mice (independent samples, 1 male mouse, 1 female mouse) separately, using the same configuration of IST-editing. As shown in newly added Fig. 9 and 10 in the appendix, IST-editing achieved consistent generalization results, reflected by the good reconstruction of unseen brain slices from a different mouse. For your further examination of generalization experiments, the mouse brain data, generated 73728 X 106496 WSIs, and updated code are accessible through our GitHub [repository](https://github.com/CTPLab/IST-editing).
>
> [1] Yao, Z., van Velthoven, C.T.J., Kunst, M. et al. A high-resolution transcriptomic and spatial atlas of cell types in the whole mouse brain. Nature 624, 317–332 (2023).
>
> ____
>
> **Q4:** how are alpha hyperparameters in Eq. 1 fixed? To which values?
>
> **A: (Prior study)** Following the recommendation by the prior SST-editing [2] study, we determine alpha_R1 and alpha_path to be 10 and 2.
> Please see also the SST-editing [code](https://github.com/CTPLab/SST-editing/blob/b3abbd430763ecda1a01b3dc8324da3c3109e289/train_style2.py#L410) for reference.
>
> [2] Jiqing Wu, Viktor H Koelzer, SST-editing: in silico spatial transcriptomic editing at single-cell resolution, Bioinformatics, Volume 40, Issue 3, March 2024.

---

> ### Comment · Reviewer_HjWd · 2024-03-22
>
> Thanks for the detailed response!
>
> Most of my concerns (Q1, Q2, and Q4) have been nicely solved. I would suggest the authors include the hyperparameter details (Q4) in the final version of the manuscript.
>
> Nevertheless, I still have some major concerns regarding Q3, which refers to the partition train/test used for evaluating the proposed approach, and the presence of one unique slide on the Xenium dataset.
>
> - As for the author's response, I get that the main experiments on the paper are carried out using the same slide for training/testing (please, correct me otherwise). This is, only results on the training subset are shown. I understand that the results of the proposed pipeline are still good compared to the SoTA approaches. Nevertheless, this important limitation of the work should be more explicitly stated in Sections  "3. Experiments" and "4. Discussion".
>
> - I would like to thank the authors for the effort of providing additional experiments on ST dataset, with separate train/test splits. Although the qualitative assessment in Figures 9 and 10 shows promising performance, I would kindly ask the authors to provide quantitative metrics for such experiments separately for train and test subsets (i.e. "Generation on trained data" and "Generation on unseen test data"). This would allow us to better assess the generalization capabilities of the proposed method.

---

> > ### Author Response · Authors · 2024-03-22
> > **Thank you for taking the time and effort in reviewing our response.**
> >
> > 1. During the weekend, we plan to include the hyperparameter details and limitation discussion in the revision.
> >
> > 2. As it requires more time to compute the quantitative metrics, such results will be provided early next week for your follow-up examination.
> >
> > Have a nice weekend.

---

> > ### Author Response · Authors · 2024-03-25
> > **Thank you again for your careful review.**
> >
> > To your remaining questions:
> >
> > 1. We added hyperparameter details in Sec. 2.2.
> > 2. We acknowledged the model limitation of training/testing on the same WSI and referred to our generalization experiments (Fig. 9 and 10) in Sec. 3.3 for addressing the issue.
> > 3. To further substantiate the generalization results and follow your suggestion,  we have included the quantitative results (PSNR and d_FID with standard deviation) in Fig. 9 and 10.
> >
> > Please check the updated manuscript for the point-by-point revision.

---

> > > ### Comment · Reviewer_HjWd · 2024-03-25
> > >
> > > Thanks for the clarifications and additional generalization results. They have nicely solved my initial concerns on the paper.

---

> > > > ### Author Response · Authors · 2024-03-26
> > > > **Thank you for the approval of our revision. If possible, we kindly ask your re-evaluation on the preliminary rating.**
> > > >
> > > > Kind regards,
> > > >
> > > > Authors of the IST-editing study

---

### Official Review · Reviewer_2uoz · 2024-02-28

**Confidence:** 3
**Preliminary Rating:** 5
**Recommendation:** Oral
**Final Rating:** 5

**Summary:**

This work introduces a method to generative whole slide images (WSI) conditioned on gene expression data. The method is based on a patch-based GAN, but can be applied in inference to produce full WSI with high resolution. The authors show the effect of changing gene expression data on the generated WSI on cell, tissue, and animal level.

**Strengths:**

The work presents strong results on a challenging problem. The gene editing results are thorough and very interesting. The authors push their method to the limit by training on small datasets to test where it begins to break down.

The authors benchmark against several other methods and demonstrate they can't produce convincing WSI.

Code is available and the authors also make model samples available for readers to examine.

**Weaknesses:**

I found the high density of the figures, with lots of sub-panels, made them difficult to read. I appreciate this in part is because the paper packs in a lot of results and there is a page limit, but I think moving some parts to supplementary would make it more readable.

**Detailed Comments:**

Confusingly the colours used for InfinityGAN in Fig 2c are different between the two panels.

The authors say their results are realistic 'as supported by expert pathologist interpretation'.  I'm not sure it's OK to make this claim without providing some evidence.

**Justification Of Final Rating:**

The authors have reasonably responded to my comments, as well as, as far as I can tell, the comments of other authors. I'm sticking with my original rating, strong accept, I think this is a nicely executed piece of work.

**Justification Of The Preliminary Rating:**

This is a thorough piece of work with impressive results, as well as strong and in some parts quite creative validation. Code and samples are made available. This work will definitely be of interest to the community.

**Questions To Address In The Rebuttal:**

It would be nice to present some evidence to support the claim that pathologists viewed the generated images as realistic.

**Special Issue:**

Yes

---

> ### Author Response · Authors · 2024-03-16
> **To Reviewer 2uoz**
>
> Thank you for your positive assessment of our work and your recommendation of our study for oral presentation.
>
> **Q1:** High density of the figures and confusingly the colours used for InfinityGAN in Fig 2c.
>
> **A:** Given the interdisciplinary nature of our study,  we aim to efficiently engage with both the DL and biomedical communities. As a result, the manuscript adopts a balanced writing style and the figure presentation adheres to conventions typically employed in biomedical publications.
>
> To improve the accessibility of these subplots, we revised Fig. 2 and clarified the color panel of Fig. 2(c) by highlighting the meaning of PSNR and FID scores. Note that the color panel is consistently used for both radar plots, i.e., larger PSNR and lower FID mean better performance.
>
> ____
>
> **Q2:** It would be nice to present some evidence to support the claim that pathologists viewed the generated images as realistic.
>
> **A: (Cell morphometrics)** Thank you for the comment. Complementary to the pathologist's review, quantitative evidence of cell morphometrics has been provided in the radar plots of Fig. 3(c). Measured with the proportional **nuclei count**, **nuclear size**, etc. between ground-truth and generated organ regions, we can see that both values are close to 1, indicating (nearly) the same amount of nuclei and the same averaged nuclear size when comparing the real and generated regions.
>
> To ensure consistency with pathologist interpretation, this study was closely supervised by an experienced board-certified pathologist (last author of this study).

---

### Official Review · Reviewer_kxJd · 2024-02-28

**Confidence:** 4
**Preliminary Rating:** 3
**Recommendation:** Poster
**Final Rating:** 4

**Summary:**

The paper introduces IST-editing, a GAN-based method that achieves seamless generation of gigapixel WSI depicting a mouse pup based on high-resolution gene expression data. Training with paired spatial gene expression and image data, this method can synthesize images at a resolution of 106496×53248 pixels and simulate morphological transitions at cellular, tissue, and organism levels.

**Strengths:**

* The authors provide the code and generated WSIs publicly for further research and validation.

* The dataset  tissue or cellular structures based on various forms of molecular data.

* The synthesis focus on high resolution instead of focusing on smaller scales or lower resolutions of existing work.

* IST-editing is notable for its efficient training and inference on consumer-grade GPUs.

**Weaknesses:**

Although the method uses well-understood biological inputs like gene expressions, the internal workings of GANs can be difficult to interpret. Understanding how changes in the input data affect the output and ensuring that the output is biologically meaningful remains a concern or need to be discussed.

As the paired data synthesis, why the most simple and straightforward pixel to pixel based synthesis validation is not used to serve as baseline?

The results in Figure 2 is bit of questionable, it might need to further elaboration.

**Detailed Comments:**

* On the phrase "To strike a balance between the generation quality and training efficiency, we employ the paired 256 × 256 gene array and 128 × 128 image in the experiments," the grammar is correct. However, regarding the concern that paired data typically have the same size, this is not always necessary, especially in cases where input and output data serve different purposes or have different inherent resolutions. In the context of machine learning, although it's common to pair higher-resolution input data with lower-resolution output targets if the task benefits from this arrangement. It would be beneficial for the authors to explain why this decision was made for clarity.

* Regarding the request for additional description of Figure 1's right gray box, it's important to provide context cues to enhance understanding.

* The small subplots adjacent to SinDDM and SinGAN raise some questions. It is unclear how these subplots relate to the overall results presented. Additionally, there is a concern that the baseline models, as evidenced by the subplots, might be or not overfit. Could the authors please clarify the purpose of these subplots and address the issue of potential overfitting in baseline models?

**Justification Of Final Rating:**

The authors have done a great job addressing my comments.

Regarding a minor point from last year's MIDL 2023, the paper by Bao et al., "Alleviating tiling effect by random walk sliding window in high-resolution histological whole slide image synthesis" (MIDL, 2023), introduces a method for alleviating stitching artifacts. The general approach of using an overlapped moving window should also be effective. While I still believe that experimenting with a pix2pix-based GAN could be worthwhile, it may be beyond the scope of this manuscript.

**Justification Of The Preliminary Rating:**

The justification is based on paper's innovative approach in using spatial transcriptomic data for synthesizing gigapixel whole slide images.

However, there remains a need for clarification regarding the pairing of different-sized gene arrays and images, and further explanation of the subplots' relevance to avoid concerns of overfitting. Enhancing the clarity of the methodology and the results' interpretability would substantially strengthen the paper's impact and credibility.

**Questions To Address In The Rebuttal:**

Please check the weakness and detailed comments section.

**Special Issue:**

No

---

> ### Author Response · Authors · 2024-03-16
> **To Reviewer kxJd**
>
> Thank you for your constructive feedback.
>
> **Q1:** It would be beneficial for the authors to explain why this decision (higher-resolution input data with lower-resolution output targets) was made for clarity.
>
> **A: (Boundary consistency)** To ensure boundary consistency between neighboring generated images, every boundary pixel at (x, y) of the image tile is obtained using gene expression values located at (x', y'), where |x' - x| <= 64  and |y' - y| <= 64. A larger 256 X 256 gene array, containing all these values, is thus necessary to generate a 128 X 128 image tile.
>
> We have added the discussion in the revision.
>
> ____
>
> **Q2**: It's important to provide context cues to enhance understanding (of the gray box in Fig. 1(right)).
>
> **A: (Reserved and removed pixel)** For clarification, we have revised annotations for gray boxes with different border colors and have updated the right plot of Fig. 1 in the revision.
>
> ____
>
> **Q3**: Could the authors please clarify the purpose of these subplots (Fig. 2) and address the issue of potential overfitting in baseline models?
>
> **A: (Upscaling challenge)**  Following the low-resolution single-image training scheme of SinGAN/SinDDM (e.g., 250 X 250 and 223 X 223), the overfitting (or faithful generation) of the input image is an expected result. Please see also natural image examples shown in both papers.
>
> Utilizing the coarse-to-fine upscaling trick introduced in both methods, we thus present unsatisfactory upscaling results (4096 X 4096, right plots) next to the faithful generation of low-res input images (left subplots) for a clear side-by-side comparison.
>
> We have clarified these subplots in the legend of Fig. 2.
>
> ____
>
> **Q4**: Understanding how changes in the input data affect the output and ensuring that the output is biologically meaningful remains a concern or need to be discussed.
>
> **A: (Colon region interpretation)** Thank you for raising the fundamental question relevant to most _in silico_ biomedical studies. To bridge the gap between algorithmic analysis and biological plausibility,  we have presented an exemplary discussion on morphological transitions of the colon region.
>
> As elaborated in lines 185-188 and the 'Colon' row in Fig. 3(b), we observe the emergence of crypt epithelial structures orchestrated by the upscaling of leading genes including Epithelial Cell Adhesion Molecule (**_EPCAM_**). As muscle-specific genes are not represented in the top colon gene sets, the outer muscle layer remains absent in the reconstruction, providing a direct and plausible link between the in silico editing results and the known biological functions of the selected genes.
>
> ____
>
> **Q5**: Why the most simple and straightforward pixel to pixel based synthesis validation is not used to serve as baseline?
>
> **A: (Translation equivariance)** Similar to many prior GAN models, the widely-used pix2pixel model (e.g., https://github.com/phillipi/pix2pix) violates the translation equivariant property. As a result, it cannot generate arbitrarily large WSIs without introducing tile stitching artifacts.

---

### Author Response · Authors · 2024-03-16
**General response**

We thank all reviewers for your careful review and positive feedback (**Oral** recommended by R2, **Poster** by R1&3). To address your questions:

  1. Detailed responses are provided and attached below your review.
  2. The revised manuscript has been uploaded and key changes are highlighted in red for clarity.
  3. The updated code for reproducing new experiments has been made publicly available for your examination.

Should you have any further questions, we are happy to answer them.

---

### Meta-Review · Area_Chair_2enp · 2024-04-03

**Recommendation:** Accept (Oral)
**Confidence:** 5

**Metareview:**

All reviewers agreed that the proposed IST-editing method based on generative adversarial learning, which demonstrates superior performance over diffusion models, is intriguing. The experimental results, along with the provided code and publicly available generated WSIs for further research and validation, were noted positively. Considering the consistent positive feedback from all reviewers and the distinctive nature of the research topic on gene editing within the MIDL community, this meta-reviewer recommends accepting the paper for an oral presentation. However, the authors are strongly encouraged to thoroughly address all comments raised by the reviewers in their final revisions.

---

> ### Author Response · Authors · 2024-04-08
> **About the possibile contribution to a MELBA Special Issue**
>
> Dear Area Chairs and Program Chairs,
>
> we thank you for your time in reading IST-editing and your approval of its distinctive nature.   Later, we are going to upload a revised final version by carefully addressing all the reviewers' feedback.
>
> Having been passionate about the potential of algorithmic gene editing within the spatial context,  we are also happy to contribute to a MELBA Special Issue (if possible).
>
> Our commitment to the _in silico_ editing is not only encouraged by one reviewer's recommendation for the MELBA submission, but also driven by our vision of simulated intervention on gene expression that leads to the phenotype of a living system (captured by multiplexed biomedical imaging).
>
> We believe such a novel perspective on biomedical imaging could be of particular interest to the MIDL community.
>
> Kind regards,
>
> Authors of IST-editing

---

### Decision · Program_Chairs · 2024-04-05

Accept (Oral)